# Review on the Biological Detoxification of Mycotoxins Using Lactic Acid Bacteria to Enhance the Sustainability of Foods Supply

**DOI:** 10.3390/molecules25112655

**Published:** 2020-06-07

**Authors:** Belal J. Muhialdin, Nazamid Saari, Anis Shobirin Meor Hussin

**Affiliations:** 1Department of Food Science, Faculty of Food Science and Technology, Universiti Putra Malaysia, 43400 UPM Serdang, Selangor, Malaysia; nazamid@upm.edu.my; 2Halal Products Research Institute, Universiti Putra Malaysia, 43400 UPM Serdang, Selangor, Malaysia; shobirin@upm.edu.my; 3Department of Food Technology, Faculty of Food Science and Technology, Universiti Putra Malaysia, 43400 UPM Serdang, Selangor, Malaysia

**Keywords:** mycotoxins, biological detoxification, LAB, sustainability, food supplies

## Abstract

The challenges to fulfill the demand for a safe food supply are dramatically increasing. Mycotoxins produced by certain fungi cause great economic loss and negative impact on the sustainability of food supplies. Moreover, the occurrence of mycotoxins at high levels in foods poses a high health threat for the consumers. Biological detoxification has exhibited a high potential to detoxify foodstuffs on a cost-effective and large scale. Lactic acid bacteria showed a good potential as an alternative strategy for the elimination of mycotoxins. The current review describes the health and economic impacts associated with mycotoxin contamination in foodstuffs. Moreover, this review highlights the biological detoxification of common food mycotoxins by lactic acid bacteria.

## 1. Introduction

Mycotoxins are the secondary metabolites of fungi that are produced in foods at ambient conditions [1]. However, only certain fungi produce mycotoxins that are hazardous to health. Significantly, fungi may produce several different mycotoxins and some mycotoxins can be produced by several fungi. The majority of mycotoxins have low molecular weight (<700 Da), are very stable in harsh conditions, and it is very difficult to eliminate them from foods and commodities [2]. There are more than 400 mycotoxins in nature, including modified mycotoxins, and they are associated with many health risks for humans and animals [3]. Therefore, authorities around the world have established limits for the maximum acceptable levels of mycotoxins associated with foods. The fungi belonging to the genera *Aspergillus*, *Penicillium*, *Fusarium,* and *Alternaria* are the most common mycotoxin producers in foods. While the most common mycotoxins found in foods are aflatoxins B1, B2, G1, and G2, ochratoxin A, fumonisins B1, B2, and B3, deoxynivalenol, zearalenone, T-2 toxin, HT-2 toxin, citrinin, ergot alkaloids, and patulin [4] (Table 1). The main target for fungal growth and mycotoxin production are foodstuffs of plant origin, including cereals, fruits, hazelnuts, almonds, seeds, and fodder [5]. On the other hand, mycotoxin residues are observed in some food products of animal origin such as milk, cheese, eggs, and meat as the result of contaminated feed [6,7]. Fungal growth and mycotoxin production can occur in foods in the field or during storage. The contamination of crops mostly originates from the field and spreads during bad handling and bad storage conditions [8]. However, mycotoxins are present in every part of the world and with a higher prevalence in tropical and subtropical regions [9,10].

The high risk associated with mycoxotins is due to their heat stability during the cooking process and the incapability of normal food procedures to remove them [11]. Moreover, the fate of degraded mycotoxins during food processing is not well known due to the limited analytical methods related to targeting contaminants [12]. However, some mycotoxins, such as deoxynivalenol, were found to partially degrade during the preparation of biscuits, and the degradation products were found to be less toxic than the parent mycotoxin, though they still remained toxic [13]. Previously, the concern about health problems associated with mycotoxins was associated with their short-term effects. Currently, the focus of research is on the long-term effects of high exposure to mycotoxins, which can cause several diseases including teratogenic diseases (disruption of the development of the embryo or fetus), carcinogenic diseases (substances causing cancer), and immune-suppressive diseases (reducing the efficacy of the immune system). Mycotoxins such as deoxynivalenol have been clearly demonstrated to inhibit protein synthesis and have a negative impact on the gastrointestinal tract due to the high exposure therein compared to other organs [14]. Furthermore, researchers observed reductions in specific pancreatic and digestion enzyme activity, including chymotrypsin, amylase, and lipase, with the consumption of high doses of aflatoxins [15]. In line with these effects, the mycotoxins deoxynivalenol and fumonisins have been reported to inhibit protein synthesis and interfere with DNA and RNA, which can cause immune suppressive diseases [16]. The distribution of enzymes, the digestion system, and the reduction in immune efficiency are associated with insufficient food digestion and absorption, loss of weight, and high susceptibility to pathogen infections [17]. On the other hand, mycotoxins have a high impact on the loss of raw materials and reduce the sustainability of food supplies. Food and Feed Safety Alerts (RASFF) rejected more than 7544 shipments from entering the EU zone because of the mycotoxin contamination between 2002 and 2011 [18]. There are no actual and precise figures on the feed and food losses due to fungal growth and mycotoxin production worldwide, but the Food and Agriculture Organization (FAO) estimated that approximately 25% of feed and food are lost annually worldwide because of fungi and the associated mycotoxins [19].

Moreover, the highest cost is associated with the detoxification of mycotoxins from raw materials because it involves the development of methods and strategies to remove the mycotoxins. According to Bata and Lásztity [9], there are three possible ways to reduce mycotoxin health hazards including prevention of contamination, prevention of the absorption of mycotoxins by the consumer’s digestive tract, and decontamination of mycotoxins from food and feed. The prevention of contamination has been successful through breeding crops to be resistant to mold infections. However, the crops are protected postharvest and the potential of contamination is very high during handling and storage. On the other hand, preventing the absorption of mycotoxins by commercially available adsorbents is a novel method to reduce the toxic impact of mycotoxins in the digestive tract [25]. The main disadvantages of this method are the high possibility of adsorbing valuable micronutrients, thus this method is more suitable for animal feeding. Another proposed novel technique is the application of fullerols C_60_(OH)_24_ (FNP) nanoparticles to modify the biosynthesis of secondary metabolites by fungi including aflatoxins [27,28]. However, the limitations of the nanoparticles technique are its dose dependence, as well as the effect being found to be limited to a short period. Decontamination is the most suitable method to reduce mycotoxin levels in foods, in which these toxins are removed from the raw material before it is consumed. There are many physical, chemical, and biological methods being used for food and feed decontamination. In addition, different approaches are used in combination in order to remove or degrade mycotoxins without impacting the quality of the raw materials. Chemical methods, including alkalization, oxidation, acidification, ammoniation, and reduction, are applied in the food and feed industry to remove or reduce the mycotoxin content [2]. Hundreds of chemicals have showed potential to reduce the mycotoxin content in foodstuffs, such as calcium hydroxide monoethylamine, ammonia, sodium hydroxide, calcium hydroxide, ozone, and chlorine. Chemical methods such as oxidation and alkalization can partially destroy mycotoxins but also destroy some of the valuable nutrients [4]. Physical methods, such as dry cleaning, milling, color sorting, irradiation, floating, washing with water, and removal of damaged grains, provide partial removal of mycotoxins [4]. However, the biological decontamination of mycotoxins was proposed as a very promising alternative with the possibility of using hundreds of microorganisms including fungi, yeast, and bacteria. With respect to all microorganisms, lactic acid bacteria (LAB) is on the top of the list of microorganisms for the degradation of mycotoxins. Therefore, the aim of this review is to highlight the high value of biological detoxification, including LAB decontamination, and show their applications in foods.

## 2. Biological Detoxification

Biological detoxification is defined as the application of microorganism enzymes and their metabolites for mycotoxin degradation. Biological detoxification is a very promising alternative, with the possibility of using hundreds and/or thousands of suitable microorganisms and metabolites. On the other hand, microbial detoxification is identified as the biotransformation of mycotoxins into less or nontoxic compounds [25]. The microorganism to be used for the detoxification should meet a certain criteria, such as being safe to use, nonpathogenic, it should produce stable known nontoxic metabolites, degrade mycotoxins, form complexes that are not revisable, be active during storage, not produce an unacceptable odor or taste, maintain the nutrient value, and have minimum requirements for its cultivation and production. Several microorganisms have been suggested to serve as detoxification agents in food and feed but only few were further investigated for practical applications because of various limitations. Fungi, yeast, and bacteria are the most common microorganisms used for the detoxification of feed and food [29]. Karlovsky [30], reviewed the potential of using several microorganisms for detoxification including *Rhizopus* sp., *Corynebacterium rubrum, Candida lipolytica, Aspergillus niger, Trichoderma viride, Mucor ambiguous, Neurospora* spp., *Armillariella tabescens,* and lactic acid bacteria. Several fungi belonging to the genus Aspergillus spp. were reported to have the ability to degrade and convert aflatoxins B1 to B2 and B3 from foodstuffs due to their enzymes [31,32]. The white rot fungi efficiently degraded aflatoxin B1 (87%) through the production of laccase enzymes in liquid media [33]. In another study, *A. niger* (ND-1) supernatant degraded 58.2% of the aflatoxins in ambient conditions [34].

In contrast, several studies reported the possibility of using certain bacteria to detoxify mycotoxins in foodstuffs [35]. *Flavobacterium aurantiacum* was one of the first used bacteria to degrade aflatoxin B1 in feed, and it was observed that the activity is related to the bacterial enzymes [36]. Sangare et al. [37] demonstrated the detoxification activity of *Pseudomonas aeruginosa* N17-1, which was able to highly degrade several aflatoxins including aflatoin B1, B2, and M1 in nutrient broth. Additionally, other bacteria species have also been reported to be able to degrade mycotoxins, such as *Bacillus subtilis* degrading aflatoxins [38], *Devosia mutans* 17-2-E-8 degrading deoxynivalenol [39], *Eubacterium* sp. BBSH797 degrading deoxynivalenol [40], soil bacterial consortium (called DX100) degrading deoxynivalenol [41], and *Lactobacillus brevis* degrading patulin [42]. In addition, certain microorganisms were reported to utilize mycotoxins as their source of carbon. The bacteria belonging to the *Agrobacterium–Rhizobium* group isolated from soil was able to convert deoxynivalenol to a less toxic metabolite named 3-keto-4-deoxynivalenol [43]. In another study, the bacterium *Devosia mutans* 17-2-E-8 degraded deoxynivalenol and the major metabolite was 3-epi-deoxynivalenol while the minor metabolite was 3-keto-deoxynivalenol, and both showed a toxicity lower than that of the parent mycotoxin [39].

The use of living cells and bioactive metabolites such as enzymes produced by certain microorganisms have a high potential for applications in the food and feed industries [44]. Some of the microorganisms are capable of degrading mycotoxins with their enzymes and use them as a carbon source. The focus of future research should consider these microorganisms during the early stage of screening for novel detoxifying microorganisms. Thus, the screening method is crucial to determine the ability of mycotoxin degradation using suitable testing conditions. Several lactic acid bacteria strains were observed to be able to degrade aflatoxin B1 and other aflatoxins due to the production of bioactive compounds [45]. Biological detoxification is a promising method that can be further improved by focusing on the production of detoxification enzymes. Hence, the isolation of suitable microorganism, optimizing the growth and production conditions, the preparation of low-cost production media, and the establishment of downstream techniques are the keys to success for the use of these enzymes in the food and feed industries. The advantages of biological mycotoxin degradation are its low cost, the broad spectrum of target mycotoxins, the minimal side effects regarding nutrients, the minimal individual training, and its suitability for a wide range of liquid and solid foods.

## 3. Lactic Acid Bacteria Detoxification Activity

Lactic acid bacteria (LAB) are on the top of the list of microorganisms for the degradation of mycotoxins due to their good safety history in food applications. LAB are preferred over other microorganisms because they are very safe for use in food, grow naturally in the human gut, which makes them function well to remove mycotoxins, and there are numerous strains that are easily cultured and maintained [32,45]. LAB has two mechanisms for the detoxification of mycotoxins from foods. Food detoxification by LAB is achieved using the viable cell of the microorganisms and/or is achieved using the enzymes produced by certain LAB strains. LAB have a long history in food preservation from spoilage microorganisms including fungi [32]. LAB produces several bioactive metabolites that can limit the growth of fungi and prevent the production of mycotoxins in food. LAB bioactive compounds include acids, carbon dioxide, hydrogen peroxide, phenyllactic acid, and bioactive low molecular weight peptides [46]. LAB produces numerous proteolytic enzymes that can hydrolyze proteins including cell-wall bound proteinase that hydrolyzes the protein into polypeptides, peptide transporters that transfer the peptides into the cell, and abundant intracellular peptidases that degrade the transferred peptides to amino acids [45]. The proteolytic enzymes of LAB play the most important role in the process of detoxification of mycotoxins in foodstuffs [47,48]. The application of LAB for mycotoxin elimination in foods has been extensively studied (Table 2). There are various advantages to using LAB cells and metabolites to remove mycotoxins from foods, and several species are generally recognized as safe (GRAS).

On the other hand, the adsorption of mycotoxins by the cell wall of LAB strains was suggested as another mechanism responsible for the removal of mycotoxins from certain foods. This activity was linked to the presence of polysaccharides, protein, and peptidoglycans in the cell wall of LAB strains [67]. Wang et al. [61] reported a reduction in patulin in cultured media due to the binding activity of selected LAB. The binding activity was increased using the strains of highest specific surface area and cell wall volume due to the high number of binding sites. The author suggested that the main functional components involved in adsorbing patulin were polysaccharides and protein. In another study, the binding of mycotoxins by LAB cells was proposed to depend on certain factors such as the initial concentration of mycotoxins, LAB cell number, LAB strain, the complexity and pH of the food, and the incubation temperature [60]. According to Dalié et al. [68], the viability of the cells was found not to be essential because the aflatoxin B1 bound to a specific monoclonal antibody found in the cell wall. Another mechanism of mycotoxin reduction in foodstuff is due to the interaction between the mycotoxins and the metabolites generated by the LAB strains such as acids, phenolic compounds, fatty acids, reuterin, and low molecular weight bioactive peptides [63]. Those metabolites can bind with the mycotoxins and may lead to a reduction in the toxicity. The mechanisms of mycotoxin degradation and removal by LAB cells and metabolites are still not fully understood, and several mechanisms have been proposed, such as degradation activity due to proteolytic enzymes and the binding of certain metabolites with the mycotoxins. However, the findings of previous studies indicated three possible mechanisms to be involved including mycotoxins degradation by LAB enzymes, adsorption by LAB cells, and the interaction of mycotoxins with LAB metabolites (Figure 1).

### 3.1. Aflatoxins Degradation by LAB

Aflatoxins comprise the major mycotoxin group produced by several species belonging to the genus *Aspergillus* including *Aspergillus flavus* or *A. parasiticus*. There are many forms of aflatoxins such as B1, B2, G1, G2, M1, and M2, but B1 is the most common in foods. The world health organization (WHO) and the International Agency for Research on Cancer (IARC) classified aflatoxins as carcinogenic to humans (Group 1) [21]. Aflatoxins are easily absorbed by the gastrointestinal tract and the absorption was estimated to be 80% higher compared to other mycotoxins [34]. Aflatoxin B1 is the most studied fungal toxin due to its high toxicity compared to the other aflatoxin forms. The mechanism of aflatoxin B1 degradation was reported to be the interaction with the gene expression of the targeted fungi. *L. brevis* demonstrated a 90–96% reduction in aflatoxin B1 produced by *Aspergillus flavus* and *Aspergillus parasiticus,* and a correlation was observed with the reduction in the *Omt-A* gene (60–70%), which is essential for the conversion of sterigmatocystin to O-methylsterigmatocystin for aflatoxin B1 biosynthesis [69]. Dairy strains including *L. rhamnosus GG* and *L. rhamnosus* LC-705 degraded 80% of aflatoxin B1 from liquid media. In addition, the degradation activity showed a strong interaction between the incubation temperature (37 °C optimum) and the bacterial cell concentration (2 × 10^9^ cfu/mL) [49]. The findings of this study demonstrated a limitation for that specific strain for food applications due to the high cell density required for detoxification. The strain *L. amylovorus* and *L. rhamnosus* degraded more than 50% of aflatoxin B1 as a result of the bacterial cell’s ability to bind to the aflatoxin [50]. This binding was found to be reversible as the aflatoxin B1 was easily released when the cells were subjected to aqueous washes. Probiotic LABs including *L. paracasei* LOCK 0920, *L. brevis* LOCK 0944, and *L. plantarum* LOCK 0945 caused a reduction in aflatoxin B1 during fermentation by 55% after 6 h of incubation at 37 °C in aerobic conditions [55]. A recent in vivo study demonstrated the ability of *L. plantarum* C88 to reduce the toxicity of aflatoxin B1 during oral administration to mice with liver oxidative damage via the inhibition of cytochrome P450 (CYP 450) 1A2 and CYP 3A4 expression [56]. In another study, a single strain (*L. plantrium*) exhibited high degradation of aflatoxin M1 when introduced into milk during yogurt production in combination with the yogurt strains (*Streptococcus thermophilus* and *Lactobacillus bulgaricus*) [53]. From the previous studies on aflatoxins, it can be concluded that several LAB isolates are able to reduce the aflatoxin content in several liquid, semiliquid, and solid food systems. Therefore, the biological detoxification activity of LAB is a promising strategy for a broad range of food applications.

### 3.2. Ochratoxin A Degradation by LAB

Ochratoxins are heat stable mycotoxins with a molecular weight of 0.4 kDa that are produced by fungi belonging to the *Aspergillus* and *Penicillium* species and they can promote tumors in humans and animals [70]. Ochratoxin A is the most toxic among this group and is associated with several health problems, including nephrotoxic, hepatotoxic, teratogenic, and carcinogenic diseases [71]. Ochratoxins A are found in numerous foods such as cereals, beer, wine, cocoa, coffee, dried vine fruit and spices, and meat products [72]. Ochratoxin A most commonly occurs in solid food and biological detoxification is very difficult. LAB strains and their metabolites showed antagonistic and synergetic degradation activity in several foodstuffs. The strain *L. acidophilus* VM 20 caused a significant reduction (95%) in ochratoxin A concentration in a liquid medium due to their antagonistic activity. The detoxification activity was further confirmed using the hepatoma cell line (HepG2), with the novel strain degrading more than 50% ochratoxin A [58]. The researchers observed that the LAB binding activity depends on the ochratoxin A concentration, LAB cell density, the pH-value, and on the viability of the LAB cells. *L. rhamnosus GG, L. acidophilus CH-5, L. plantarum BS, L. sanfranciscensis,* and *L. brevis* showed a 50% reduction in ochratoxin A when tested in liquid medium, but the binding of ochratoxin A with LAB cells was observed to be reversible [73]. An in vitro evaluation of the detoxification activity of the LAB strains, including *L. plantarum*, *L. brevis*, and *L. sanfranciscensis,* demonstrated a 50% reduction in ochratoxin A after incubation for 30 min in 1 M phosphate buffer (sodium acetate 0.615%, EDTA 0.1%, MgCl_2_ 0.254%, raffinose 29.72%, pH 6.2). The researchers reported that ochratoxin A was adsorbed by the thermally inactivated LAB cells to such as degree due to the increase in hydrophobicity [74]. Abrunhosa et al. [59] determined the detoxification activity of *Pediococcus parvulus* towards ochratoxin A in a liquid system. Under optimum conditions (incubation at 30 °C for 7 days), 90% of ochratoxin A was degraded in 19 h due to the conversion of OTA into OTα. The degradation activity showed a strong correlation with the inoculum size and the incubation temperature. A recent study demonstrated the advantage of combining several LAB strains with *Saccharomyces cerevisiae* as a symbiotic system to enhance the degradation activity of ochratoxin A. The optimum combination showed a reduction in ochratoxin A in the range 31.9–47.7% after 24 h of incubation at 30 or 37 °C. In addition, the in vivo experiment showed that the ochratoxin was reduced in the contaminated feed to a level below 2 mg kg^−1^ [75].

### 3.3. Patulin Degradation by LAB

Patulin is a water-soluble mycotoxin produced by several fungi belonging to the genera *Aspergillus* and *Penicillium*. However, *Penicillium expansum,* which is known as the apple rot fungi, is the most common species that is associated with patulin production [76]. It occurs in a diverse range of foodstuffs, such as grapes, apples, peaches, olives, pears, and cereals [77,78]. Patulin residues can cause specific safety issues because it has the ability to penetrate the body tissues and inhibit protein synthesis, which results in the reduction in glycogen concentration in the liver, kidney, and intestinal tissues [79]. Patulin is hypothesized to cause cancer for humans and animals, though International Agency for Research on Cancer (IARC) did not have enough evidence to prove this hypothesis [42]. There are very limited studies about the biodegradation of patulin by LAB strains. *Bifidobacterium animalis* VM degrade 80% of patulin, and the activity was observed to be related to the patulin concentration in the media and the LAB cell density. The detoxification activity was further confirmed using the human hepatoma cell line (HepG2), which showed an improved division rate in the presence of the selected LAB [58]. The heat inactivated *L. brevis* 20,023 showed a high adsorption ability towards patulin into the cell wall of the selected LAB in a liquid medium. The presence of functional groups, including polysaccharides and proteins, was observed to significantly increase the adsorption of patulin into the LAB cells [61]. In an earlier study, the LAB strains *Bifidobacterium bifidum* 6071 and *L. rhamnosus* 6149 were able to degrade patulin in a liquid system and the percentage was 52.9% for viable and 54.1% for nonviable cells at pH 4 and an incubation temperature of 37 °C [80]. Patulin can be greatly degraded at low pH levels and this may be the main factor in the LAB degradation activity. In a very recent study, chemical degradation of patulin in the presence of ascorbic acid apple juice was evaluated. The patulin was reduced by 60% in the apple juice mixed with 0.25% (*w*/*v*) ascorbic acid, and the biodegradation led to the production of less-toxic metabolites [81]. The author suggested that the mechanism was due to the degradation of patulin into a less toxic compound known as (E/Z)-ascladiol (ASC-E/Z).

## 4. Application of LAB in Foods

The mycotoxins degradation potential of LAB strains has made them one of the most important candidates for the detoxification activity in foodstuffs. This potential in food applications is due to the properties of LAB, which include the generally recognized as safe (GRAS) status, the low production cost, and the broad spectrum of food applications. Moreover, LAB strains can reduce the economic losses associated with foodstuffs by extending their shelf life and preventing mycotoxin production [80]. The applications are divided into the use of the live LAB cells and/or the use of their enzymes and metabolites. Fermented milk was prepared using *L. casei* LC-01, showing a significant reduction (50–58%) of aflatoxin M1 in milk after 21 days of storage at 4–6 °C [82]. In another study, kefir grains, which contain a diversity of LAB strains, demonstrated the ability to reduce ochratoxin A in milk by as much as 81% [83]. LAB play a very important role in bakery products such as sourdough and fermented cereal products. Several LAB strains including *L. plantarum* were reported to reduce 84.1–99.9% of the aflatoxin B1 in bread [84]. LAB were applied to degrade the patulin in contaminated apple juice and they efficiently reduced patulin content by 80% without any effects on the apple juice quality [80]. Previous studies demonstrated the high potential of selected LAB strains in the elimination of several mycotoxins in foodstuffs, which will enhance the sustainability of the supply chain. LAB applications are suitable to degrade mycotoxins in a broad range of foodstuffs due to their strong enzymatic system and fast adaptation to several substrates. The application of LAB was recently recommended as a competent strategy to remove and/or degrade mycotoxins in foodstuffs [85]. LAB strains can be applied in foodstuffs as a co-culture, such as in the hurdle strategy that is used to remove mycotoxins in dairy products [86]. In addition, whey permeate cheese fermented with LAB containing several metabolites was reported as an effective strategy for the reduction of mycotoxin produced by *Fusarium* spp. in malting wheat grains used to produce beverages and bakery products [87]. LAB fermentation can significantly reduce mycotoxin content in foodstuff. Moreover, fermented foodstuffs can be transformed into other processed food products to reduce the loss caused by mycotoxin contamination.

## 5. Conclusions and Future Study

Mycotoxins produced by several fungi in foodstuffs are the source of a very serious health threat to consumers. Moreover, they have a very strong impact on the sustainability of food supplies due to losses associated with plant-based foodstuffs. Biological degradation of mycotoxins has a great potential to reduce the health hazards and economic losses. Certain lactic acid bacteria strains were found to be able to remove different mycotoxins from foodstuffs by binding to their cell wall or by degradation with their enzymes. The advantages of using LAB strains for mycotoxin detoxification includes their direct application in foods, the broad spectrum of their activity, and the low cost of the detoxification process. However, more effort is needed to determine the mechanisms of mycotoxin degradation and the optimum conditions for the detoxification. The detoxification activity of LAB is strain-dependent and there is a need to determine LAB diversity and suitability for different food applications. Future studies should focus on combining selected LAB strains that have different detoxification mechanisms to improve the efficiency of mycotoxin degradation.

## Figures and Tables

**Figure 1 molecules-25-02655-f001:**
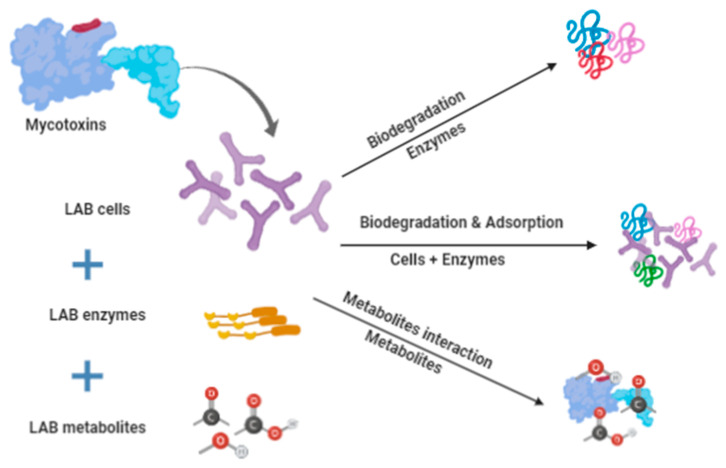
Diagram of the proposed mechanisms responsible for mycotoxin degradation by lactic acid bacteria strains.

**Table 1 molecules-25-02655-t001:** Food related mycotoxins and fungi.

Mycotoxin	Fungi	Target Foods	Reference
Dioxynivalenol	*Fusarium graminearum*	Cereal products	[20]
Aflatoxins	*Aspergillus flavus, A. nomius, A. parasiticus*	Wheat, corn, rice, peanut, rice, pepper, cotton, nut tree and spices	[21]
Fumonisins	*F. proliferatum*, *F. verticillioides*, *F. nygamai*	Rice, figs, beer and corn	[22]
HT-2 Toxins	*Fusarium* spp.	Oat, corn	[23]
Ochratoxin	*A. niger, A. ochraceus, A. carbonarius, Penicillium verrucosum*, *Neopetromyces* spp., *Petromyces* spp.	Fruits, coffee, spices, wine, dried cocoa, beans, corn, cereal, grains, and rice.	[24]
Patulin	*P. patulum, P. expansum, P. urticae, A. terreus, A. clavatus, Byssochlamys nivea, P. patulin*	Apricots, grapes, grape fruit, peaches, pears, apples, fruit juice, meat, cheese and cereals	[25]
T-2 toxin	*F. sporotrichioides, F. poae, F. equiseti*, *F. acuminatum*	Wheat, corn, oats, barley, rice, beans, and soya beans	[26]
Zearalenone	*F. graminearum, F. culmorum, F. cerealis, F. equiseti, F. crookwellense, F. semitectum*	Cereal products, wheat, barley, oat, soybean and corn	[23]

**Table 2 molecules-25-02655-t002:** Lactic acid bacteria reported applications for mycotoxins degradation in foods and feeds.

Microorganism	Target Mycotoxin	Degradation%	Reference
*Lactobacillus rhamnosus GG*, *L. rhamnosus* LC-705	Aflatoxin B1	80%	[49]
*L. amylovorus, L. rhamnosus*	Aflatoxin B1	50%	[50]
*L. casei* LOCK 0920, *L. brevis* LOCK 0944, *L. plantarum* LOCK 0945	Aflatoxines (Bl, B2, G1, G2)	~50%	[51]
*L. plantarum, Lactococcus lactis*	Aflatoxin B1	81%	[52]
*Streptococcus thermophiles, L. bulgaricus, L. plantrium*	Aflatoxin M1	11–34%	[53]
*L. casei*	Aflatoxin B1	49.2%	[54]
*L. paracasei* LOCK 0920, *L. brevis* LOCK 0944, *L. plantarum* LOCK 0945	Aflatoxin B1	39–55%	[55]
*L. plantarum* C88	Aflatoxin B1	60%	[56]
Lactic acid bacteria strains	Aflatoxins B1 and B2	ND	[57]
*L. casei* LOCK 0920, *L. brevis* LOCK 0944, *L. plantarum* LOCK 0945	Ochratoxin A	~50%	[51]
*L. acidophilus* VM 20, *Bifidobacterium animalis* VM12	Ochratoxin A Patulin	95% 80%	[58]
*Pediococcus parvulus*	Ochratoxin A	90%	[59]
*L. rhamnosus* CECT 278T	Ochratoxin A	97%	[60]
*L. brevis* 20023	Patulin	ND	[61]
*L. plantarum* GT III	Deoxynivalenol	56–66%	[62]
Lactic acid bacteria strains	Deoxynivalenol, fumonisins B1, fumonisins B2	55%, 82%, and 100%	[63]
Lactic acid bacteria	Fumonisin B1 Zearalenone	56–67% 68–75%	[64]
*L. rhamnosus GG* (ATCC 53103), *L. plantarum* A1	Zearalenone	ND	[65]
*L. paracasei, L. lactis*	Zearalenone	55%	[66]

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
