# Peer review of "Review on the Biological Detoxification of Mycotoxins Using Lactic Acid Bacteria to Enhance the Sustainability of Foods Supply"

_molecules, 2020, doi:10.3390/molecules25112655_

Round 1
Reviewer 1 Report
The manuscript reviews the usage of Lactic acid bacteria as mycotoxins detoxifying agents. Subject is very interesting in the field, and here are suggestions to make the review better:
General comments: in the text please use exact numbers and metabolites, otherwise it sounds non-scientific. More references should be included in review type of paper.
Line 5 please use affiliations by the order (Belal J Muhialdin1,2 instead1,3), be careful to change affiliations in the list (line 6-11).
Line 10 please add postal number (full postal address, as with other affiliations)
Line 12 please add word “safe” between “for” and “food”
Line 25 remove word “some”, and add “different” between “several” and “mycotoxins”, and replace “some” with “same”
Line 29 please add that the modified mycotoxins are also problematic
Line 31 please add also the Alternaria genera as major producers of mycotoxins ( https://doi.org/10.1016/j.cofs.2017.09.012)
Line 33 please add T-2 toxin, HT-2 toxin, citrinin, ergot alkaloids
Line 39 – use word “improper” instead of “bad”
Line 40 – please add also following references with high mycotoxin results, co-occurrences of multiple mycotoxins, and risk assessment: ( https://doi.org/10.1016/j.fct.2018.08.025 )
Line 42 – please comment also the modifications of the mycotoxins during heat processing (book: ILSI Europe PRACTICAL GUIDANCE TO MITIGATION OF MYCOTOXINS DURING FOOD PROCESSING; and other publications: (https://doi.org/10.1007/s00216-019-02101-9; https://doi.org/10.1016/j.foodchem.2018.11.150; https://dx.doi.org/10.3390%2Ftoxins11060317 )
Lines 46 – 49: please specify for each of toxic effect which of the mycotoxins are main causative agents.
Line 49 – 51 – please define which mycotoxins cause reductions of specific enzymes, and define which enzymes.
Line 52 – please specify the mycotoxin that is causing those effects, and add also that there can be other modified varieties of the same mycotoxin
Line 56 – EFSA is not authority that’s banning import of food and feed, EFSA is just giving opinions to legislation of European union. Please re-check all of the used numbers and update with RASFF portal
Table 1 : please add also HT-2 toxin, and citrinin, and arrange them alphabetically; use small letter for (sub)species name (A. nomius)
Line 73: change word “bad” with “toxic”
Line 74: please comment also other used approaches: usage of additives and nanoparticles (https://doi.org/10.1111/gfs.12452; https://doi.org/10.1007/s10532-010-9371-y; https://doi.org/10.3382/japr.2010-00168; https://doi.org/10.1038/s41598-020-57706-3; https://doi.org/10.1038/s41598-018-31305-9; https://doi.org/10.1002/mnfr.200500181 )
Line 78 – please support your claim with strong new reference, and clarify what do you mean as a single method for detoxifying (one method for all toxins, or all available methods for one (myco)toxin)?
Line 86 – please give several examples in references where physical methods did not succeed in removing (Review here is claiming differently: https://dx.doi.org/10.1016%2Fj.foodcont.2017.01.008 ; http://ijhse.ir/index.php/IJHSE/article/view/136; https://doi.org/10.3920/WMJ2014.1766 )
Line 87-94: please support your claims with strong references, or rephrase it
Line 96 – please add “, enzymes” between “microorganism” and “and”
Line 99 – please add “is” between “detoxification” and “identified”
Line 101 – please add “known” between “stable” and “nontoxic”
Line 103 – please add reference at the end of the sentence (Starting in line 100)
Line 110 – please change word “but” into “and”, and specify what do you consider good and acceptable. This is quite subjective, please try to be more objective and specific – declare exact reduction and detoxified products. Considered this in all future mentioning of the detoxified products – please define what were they, if they were not classified, confirmed and named, please mention also that. Those are important information, especially since there ware may declared detoxified products that had changed structure and could not be detected, but the toxicity remained (eg. DON was “detoxified” to DON-3-Glucoside, that was still toxic to humans since our gut microbiota cleaved off the Glucoside, and we absorbed native toxin- DON).
Line 112 – please specify mycotoxin detoxifying metabolites
Line 114 – define to what metabolite was AFT degraded?
Line 120 – 123 – define degradation products
Line 126 – please add reference for some microorganisms that use mycotoxins as carbon source, and give exact examples.
Line 138 – please mention also other mycotoxins (e.g.: https://doi.org/10.3920/WMJ2014.1807; https://doi.org/10.1007/s10532-010-9371-y; https://dx.doi.org/10.1128%2FAEM.01438-09; https://doi.org/10.1016/j.jbiotec.2009.11.004; https://doi.org/10.2520/myco1975.2006.Suppl4_155; https://dx.doi.org/10.3390%2Ftoxins11090523; https://dx.doi.org/10.1038%2Fsrep29105; https://doi.org/10.3920/WMJ2015.2005; https://doi.org/10.3382/ps/pew052 )
Line 140 – 141 – please add reference
Line 141-142 – please rephrase second sentence (… it has very safe use in food…), and add supporting reference
Line 143 – please change “paths” with “mechanisms”
Line 145-6: use only LAB, you have already defined abb in the first sentence of the paragraph, also please add supporting reference
Line 149 – bacteriocins are usually effective against bacteria, not fungi.
Table 2 – please group the degradations by the mycotoxin, and add detoxified product column where you can explain from which mycotoxin, by which LAB you get what degradation product.
Line 162 -167 – please add that also other live or death microbes (yeast, or yeast cell wall) can also be used for mycotoxin binding / detoxification (https://doi.org/10.1078/0723202042369947)
Line 172 – please define mechanism and give examples
Figure 1 – please change the figure of mycotoxins – this form is usually used for the proteins/enzymes/antibodies, and in the figure the mycotoxins look larger than LAB cells/enzymes/metabolites
Line 184 – please change the reference to other dealing with adsorption of aflatoxins (e.g : https://dx.doi.org/10.1093%2Ftoxsci%2Fkfq283)
Line 188 – please define what strain of Aspergillus spp is producing AFB1
Line 189 – please define what is Omt-A gene used for in aflatoxins biosynthesis
Line 191 – please define the degradation products
Line 192 – please define were it is applicable to have such huge number o LAB in food?
Line 194 – please define also in others if the binding was reversible, and specify under which conditions it can be reversed.
Line 198 – by inhibiting P450 you will reduce the toxicity of AFB1, but not other mycotoxins that don’t need activation, and use P450 as first step of detoxification.
Line 203 – please define the effectiveness of LAB without mixing with S. cerevisiae, otherwise it is not the LAB effect.
Line 216-217 – please explain degradation mechanism and degradation products
Line 220 – please explain type of dependence (antagonistic, agonistic, synergistic..)
Line 226 – please define temperature, pH, molality, and reduction metabolites
Line 229 – 231 – define the tested temperatures – degradation activity is higher at higher temperature (tested interval form x to y °C)
Line 233 – the combination of LAB and S. cerevisiae, is less effective (32%) than LAB alone (90%)?
Line 243 change to IARC
Line 256 – add T (The highest Patulin), and specify mechanism
Line 257 – please add also other important mycotoxins from table 1
Line 259 change word “ideal” with “one of”
Line 263 – please add reference where LAB are extending the shelf life of food
Line 265 – give example where you ferment milk for 21 day, or where this LAB can be applied?
Line 269 – reduction of 99.9% is combined effect of LAB and yeast, and starch trapping
Line 275 – 276 – in the reference they urge to develop most cost –effective strategy, and do not explicitly recommend LAB.
Lin 279 change to “whey permeate cheese fermented with LAB…
Line 291 remove short incubation time, since fermentations can take up to 21 day accoding to text.
Line 294 – 296 – there are several already applied enzymes ad detoxification agents by Biomin
Author Response
The manuscript reviews the usage of Lactic acid bacteria as mycotoxins detoxifying agents. Subject is very interesting in the field, and here are suggestions to make the review better:
General comments: in the text please use exact numbers and metabolites, otherwise it sounds non-scientific. More references should be included in review type of paper.
Reply: The authors have amended the manuscript according to the suggestions received from the respected reviewer. The amendments was highlighted in red colour font all through the manuscript.
Line 5 please use affiliations by the order (Belal J Muhialdin1,2 instead1,3), be careful to change affiliations in the list (line 6-11).
Reply: The affiliation was amended according to the recommendation of the respected reviewer.
Line 10 please add postal number (full postal address, as with other affiliations)
Reply: The postal number added as recommended.
Line 12 please add word “safe” between “for” and “food”
Reply: the word “safe” added as recommended by the respected reviewer.
Line 25 remove word “some”, and add “different” between “several” and “mycotoxins”, and replace “some” with “same”
Reply: the word “some” was removed, the word “different” was added as recommended by the respected reviewer.
Line 29 please add that the modified mycotoxins are also problematic
Reply: The modified mycotoxins was added as recommended by the respected reviewer.
Line 31 please add also the Alternaria genera as major producers of mycotoxins (https://doi.org/10.1016/j.cofs.2017.09.012)
Reply: The genera Alternaria was added to the sentence.
Line 33 please add T-2 toxin, HT-2 toxin, citrinin, ergot alkaloids
Reply: The highlighted mycotoxins were added and the changes highlighted in red colour font.
Line 39 – use word “improper” instead of “bad”
Reply: The word “improper” was replaced with “bad” as recommended by the respected reviewer.
Line 40 – please add also following references with high mycotoxin results, co-occurrences of multiple mycotoxins, and risk assessment: ( https://doi.org/10.1016/j.fct.2018.08.025 )
Reply: The reference was added according to the recommendation of the respected reviewer.
Ojuri, O. T., Ezekiel, C. N., Sulyok, M., Ezeokoli, O. T., Oyedele, O. A., Ayeni, K. I., ... & Nwangburuka, C. C. (2018). Assessing the mycotoxicological risk from consumption of complementary foods by infants and young children in Nigeria. Food and chemical toxicology, 121, 37-50.
Line 42 – please comment also the modifications of the mycotoxins during heat processing (book: ILSI Europe PRACTICAL GUIDANCE TO MITIGATION OF MYCOTOXINS DURING FOOD PROCESSING; and other publications: (https://doi.org/10.1007/s00216-019-02101-9; https://doi.org/10.1016/j.foodchem.2018.11.150; https://dx.doi.org/10.3390%2Ftoxins11060317)
Reply: The references suggested by the respected reviewer was added to the introduction and highlighted in red colour font.
Stadler, D., Berthiller, F., Suman, M., Schuhmacher, R., & Krska, R. (2020). Novel analytical methods to study the fate of mycotoxins during thermal food processing. Analytical and Bioanalytical Chemistry, 412(1), 9-16.
Stadler, D., Lambertini, F., Bueschl, C., Wiesenberger, G., Hametner, C., Schwartz-Zimmermann, H., ... & Suman, M. (2019). Untargeted LC–MS based 13C labelling provides a full mass balance of deoxynivalenol and its degradation products formed during baking of crackers, biscuits and bread. Food chemistry, 279, 303-311.
Lines 46 – 49: please specify for each of toxic effect which of the mycotoxins are main causative agents.
Reply: The causative mycotoxin for each disease was stated as recommended by the respected reviewer.
Line 49 – 51 – please define which mycotoxins cause reductions of specific enzymes, and define which enzymes.
Reply: The specific enzymes activity such as chymotrypsin, amylase and lipase which was stated in the sentence.
Line 52 – please specify the mycotoxin that is causing those effects, and add also that there can be other modified varieties of the same mycotoxin
Reply: The mycotoxins causing certain effect was stated as recommended by the respected reviewer.
Line 56 – EFSA is not authority that’s banning import of food and feed, EFSA is just giving opinions to legislation of European union. Please re-check all of the used numbers and update with RASFF portal
Reply: The information was replaced with those present in the RASFF official report and the reference was added.
RASFF. (2011). The Rapid Alert System for Food and Feed‐2011 Annual Report.
Table 1 : please add also HT-2 toxin, and citrinin, and arrange them alphabetically; use small letter for (sub)species name (A. nomius)
Reply: The mycotoxin HT-2 toxin was added to table 1 as recommended but citrinin was not included due to limited information in the literature for food contamination, table 1 was re-arranged in alphabetic order.
Lattanzio, V. M., & Pascale, M. (2017). Determination of T-2 and HT-2 Toxins in Oats and Oat-Based Breakfast Cereals by Liquid-Chromatography Tandem Mass Spectrometry. In Oat (pp. 127-136). Humana Press, New York, NY.
Line 73: change word “bad” with “toxic”
Reply: The word “bad” was replaced with “toxic” as recommended by the respected reviewer
Line 74: please comment also other used approaches: usage of additives and nanoparticles (https://doi.org/10.1111/gfs.12452; https://doi.org/10.1007/s10532-010-9371-y; https://doi.org/10.3382/japr.2010-00168; https://doi.org/10.1038/s41598-020-57706-3; https://doi.org/10.1038/s41598-018-31305-9; https://doi.org/10.1002/mnfr.200500181)
Reply: Some of the relevant studies was added to the section as recommended by the respected reviewer.
Tihomir, K., Ivana, B., Marija, K., Ante, L., Frane, Č. K., Aleksandar, D., ... & Bojan, Š. (2020). Impact of fullerol C 60 (OH) 24 nanoparticles on the production of emerging toxins by Aspergillus flavus. Scientific Reports, 10(1).
Kovač, T., Borišev, I., Crevar, B., Kenjerić, F. Č., Kovač, M., Strelec, I., ... & Šarkanj, B. (2018). Fullerol C 60 (OH) 24 nanoparticles modulate aflatoxin B 1 biosynthesis in Aspergillus flavus. Scientific reports, 8(1), 1-8.
Line 78 – please support your claim with strong new reference, and clarify what do you mean as a single method for detoxifying (one method for all toxins, or all available methods for one (myco)toxin)?
Reply: The authors mean that there is no single method can detoxify all types of mycotoxins. The sentence was removed from this section to avoid confusion.
Line 86 – please give several examples in references where physical methods did not succeed in removing (Review here is claiming differently: https://dx.doi.org/10.1016%2Fj.foodcont.2017.01.008; http://ijhse.ir/index.php/IJHSE/article/view/136; https://doi.org/10.3920/WMJ2014.1766)
Reply: The statement regarding the limitation of the physical method was removed from the section.
Line 87-94: please support your claims with strong references, or rephrase it
Reply: The proper references were added to the section according to the recommendation of the respected reviewer.
Line 96 – please add “, enzymes” between “microorganism” and “and”
Reply: The word “enzymes” was added as recommended by the respected reviewer
Line 99 – please add “is” between “detoxification” and “identified”
Reply: “is” was added as recommended by the respected reviewer
Line 101 – please add “known” between “stable” and “nontoxic”
Reply: “know’ was added to the text as recommended by the respected reviewer
Line 103 – please add reference at the end of the sentence (Starting in line 100)
Reply: The reference is added as recommended by the respected reviewer
Line 110 – please change word “but” into “and”, and specify what do you consider good and acceptable. This is quite subjective, please try to be more objective and specific – declare exact reduction and detoxified products. Considered this in all future mentioning of the detoxified products – please define what were they, if they were not classified, confirmed and named, please mention also that. Those are important information, especially since there ware may declared detoxified products that had changed structure and could not be detected, but the toxicity remained (eg. DON was “detoxified” to DON-3-Glucoside, that was still toxic to humans since our gut microbiota cleaved off the Glucoside, and we absorbed native toxin- DON).
Reply: The authors decided to remove the sentence to avoid the confusion and to avoid making general statement which is very difficult to be determined if is good and/or acceptable level of decontamination.
Line 112 – please specify mycotoxin detoxifying metabolites
Reply: The mycotoxin was specified as recommended by the respected reviewer
Line 114 – define to what metabolite was AFT degraded?
Reply: The metabolites were defined in the text as recommended by the respected reviewer
Line 120 – 123 – define degradation products
Reply: The degraded mycotoxins were present in the text
Line 126 – please add reference for some microorganisms that use mycotoxins as carbon source, and give exact examples.
Reply: The relevant studies was added to the section as recommended by the respected reviwer
Shima, J., Takase, S., Takahashi, Y., Iwai, Y., Fujimoto, H., Yamazaki, M., & Ochi, K. (1997). Novel detoxification of the trichothecene mycotoxin deoxynivalenol by a soil bacterium isolated by enrichment culture. Applied and Environmental Microbiology, 63(10), 3825-3830.
Line 138 – please mention also other mycotoxins (e.g.: https://doi.org/10.3920/WMJ2014.1807; https://doi.org/10.1007/s10532-010-9371-y; https://dx.doi.org/10.1128%2FAEM.01438-09; https://doi.org/10.1016/j.jbiotec.2009.11.004; https://doi.org/10.2520/myco1975.2006.Suppl4_155; https://dx.doi.org/10.3390%2Ftoxins11090523; https://dx.doi.org/10.1038%2Fsrep29105; https://doi.org/10.3920/WMJ2015.2005; https://doi.org/10.3382/ps/pew052)
Reply: The other mycotoxins was not considered as they were not discussed in the LAB detoxification section due to limited studies on LAB interaction with those mycotoxins
Line 140 – 141 – please add reference
Reply: The references added as recommended by the respected reviewer
Line 141-142 – please rephrase second sentence (… it has very safe use in food…), and add supporting reference
Reply: The sentence was added to the text as recommended by the respected reviewer
Line 143 – please change “paths” with “mechanisms”
Reply: “path” was replaced with “mechanisms” as recommended by the respected reviewer
Line 145-6: use only LAB, you have already defined abb in the first sentence of the paragraph, also please add supporting reference
Reply: The lactic acid bacteria was abbreviated to LAB and the references was added as recommended by the respected reviewer
Line 149 – bacteriocins are usually effective against bacteria, not fungi.
Reply: Bacteriocins was removed from the sentence as recommended by the respected reviewer
Table 2 – please group the degradations by the mycotoxin, and add detoxified product column where you can explain from which mycotoxin, by which LAB you get what degradation product.
Reply: The mycotoxins in table 2 were grouped as recommended by the respected reviewer. Thus, the detoxify products was not added as the majority of the studies did not determine the by-products but only the reduction percentage.
Line 162 -167 – please add that also other live or death microbes (yeast, or yeast cell wall) can also be used for mycotoxin binding / detoxification (https://doi.org/10.1078/0723202042369947)
Reply: The authors totally agree with the reviewer that yeast cells can bind to different mycotoxins and has high potential to lower mycotoxins levels as demonstrated in several studies. However, the focus of the current review paper is lactic acid bacteria and therefore the yeast was not considered in details
Line 172 – please define mechanism and give examples
Reply: The suggested mechanism was added with the reference as recommended by the respected reviewer.
Dalié, D. K. D., Deschamps, A. M., & Richard-Forget, F. (2010). Lactic acid bacteria–Potential for control of mould growth and mycotoxins: A review. Food Control, 21(4), 370-380.
Figure 1 – please change the figure of mycotoxins – this form is usually used for the proteins/enzymes/antibodies, and in the figure the mycotoxins look larger than LAB cells/enzymes/metabolites
Reply: The figure is to illustrate the mechanisms of the LAB detoxification activity which was labelled. The figure was prepared using online software and has limitation of the illustration pictures
Line 184 – please change the reference to other dealing with adsorption of aflatoxins (e.g : https://dx.doi.org/10.1093%2Ftoxsci%2Fkfq283)
Reply: The reference replaced with the reference suggested by the respected reviewer
Kensler, T. W., Roebuck, B. D., Wogan, G. N., & Groopman, J. D. (2011). Aflatoxin: a 50-year odyssey of mechanistic and translational toxicology. Toxicological sciences, 120(suppl_1), S28-S48.
Line 188 – please define what strain of Aspergillus spp is producing AFB1
Reply: The specific species (Aspergillus flavus and Aspergillus parasiticus) were included as recommended by the respected reviewer
Line 189 – please define what is Omt-A gene used for in aflatoxins biosynthesis
Reply: The function of the gene is added to the text as recommended by the respected reviewer
Line 191 – please define the degradation products
Reply: The author did not determine the by-products of the degradation process of aflatoxin B1
Line 192 – please define were it is applicable to have such huge number o LAB in food?
Reply: The required cell density has limitation for food application, the limitation was added in a sentence following that study findings and highlighted in red colour font
Line 194 – please define also in others if the binding was reversible, and specify under which conditions it can be reversed.
Reply: The binding limitation and the release of aflatoxins B1 due to washing was added in a sentence following the finding stated
Line 198 – by inhibiting P450 you will reduce the toxicity of AFB1, but not other mycotoxins that don’t need activation, and use P450 as first step of detoxification.
Reply: The statement was amended to describe aflatoxins only and other mycotoxins were excluded
Line 203 – please define the effectiveness of LAB without mixing with S. cerevisiae, otherwise it is not the LAB effect.
Reply: The statement was removed from the section to focus on LAB cells only
Line 216-217 – please explain degradation mechanism and degradation products
Reply: The author did not determine the degradation products and the mechanism but indicated the high potential to apply LAB in food to reduce the occurrence of aflatoxins
Line 220 – please explain type of dependence (antagonistic, agonistic, synergistic..)
Reply: Due to antagonistic activity of the selected strain, the details was included in the sentence
Line 226 – please define temperature, pH, molality, and reduction metabolites
Reply: The molarity and pH of the buffer was stated as recommended by the respected reviewer. However, the author did not declare the reduction metabolites
Line 229 – 231 – define the tested temperatures – degradation activity is higher at higher temperature (tested interval form x to y °C)
Reply: The optimum conditions were added to the sentence as recommended by the respected reviewer
Line 233 – the combination of LAB and S. cerevisiae, is less effective (32%) than LAB alone (90%)?
Reply: The results of the study were explained in details and the sentence was amended and highlighted in red colour font
Line 243 change to IARC
Reply: Changed as recommended by the respected reviewer
Line 256 – add T (The highest Patulin), and specify mechanism
Reply: The T was added and the mechanism was included as recommended by the respected reviewer
Line 257 – please add also other important mycotoxins from table 1
Reply: The ability of lactic acid bacteria was not extensively studied against the other mycotoxins only few studies. However, there are several studies related to other bacteria and microorganisms but the focus of this review is on lactic acid bacteria detoxification activity
Line 259 change word “ideal” with “one of”
Reply: “ideal” was changed to “one of” according to the respected reviewer recommendation
Line 263 – please add reference where LAB are extending the shelf life of food
Reply: The reference was added to the sentence as recommended by the respected reviewer
Line 265 – give example where you ferment milk for 21 day, or where this LAB can be applied?
Reply: 21 days of storage not incubation, the sentence was extensively revised and amended
Line 269 – reduction of 99.9% is combined effect of LAB and yeast, and starch trapping
Reply: The sentence was amended according to the reference
Line 275 – 276 – in the reference they urge to develop most cost –effective strategy, and do not explicitly recommend LAB.
Reply: The sentence was amended, and the reference was replaced with relevant reference
Lin 279 change to “whey permeate cheese fermented with LAB…
Reply: The sentence was changed as recommended by the respected reviewer
Line 291 remove short incubation time, since fermentations can take up to 21 day accoding to text.
Reply: Removed as recommended by the respected reviewer
Line 294 – 296 – there are several already applied enzymes ad detoxification agents by Biomin
Reply: The sentence was removed from the recommendation
Reviewer 2 Report
In this review, the authors focus on mycotoxins and their impact on the food supply. They also detail Biological Detoxification by Lactic Acid Bacteria. The review is rather timely and fits an area of need. While there needs to be some language editing and limited changes, it is suggested this be published after minor revisions.
Table 1: The full name of the species should be listed before abbreviation. For example, Fusarium should be spelled out in the first row (instead of the abbreviated F. Graminearum) and abbreviated in the 4th row (should appear as F. proliferatum).
Section 2: Biological Detoxification
This section refers to biological detoxification as a promising alternative. However, it never mentions what the other alternatives are. To provide a broader scope and better establish the place for this review, some mention of the existing alternatives and what the current industry standard is (if any) should be included.
Author Response
In this review, the authors focus on mycotoxins and their impact on the food supply. They also detail Biological Detoxification by Lactic Acid Bacteria. The review is rather timely and fits an area of need. While there needs to be some language editing and limited changes, it is suggested this be published after minor revisions.
Table 1: The full name of the species should be listed before abbreviation. For example, Fusarium should be spelled out in the first row (instead of the abbreviated F. Graminearum) and abbreviated in the 4th row (should appear as F. proliferatum).
Reply: Table 1 was extensively amended as recommended by the respected reviewer
Section 2: Biological Detoxification
This section refers to biological detoxification as a promising alternative. However, it never mentions what the other alternatives are. To provide a broader scope and better establish the place for this review, some mention of the existing alternatives and what the current industry standard is (if any) should be included.
Reply: The alternative detoxification techniques such as chemical and physical methods is stated in the introduction lines 90-100
Round 2
Reviewer 1 Report
Thank you for the reviewed manuscript, it is much easier to follow now.
Please just update the reference in line 34 with https://doi.org/10.1016/j.cofs.2017.09.012, since you added data without reference support.
before it was: Line 31 please add also the Alternaria genera as major producers of mycotoxins (https://doi.org/10.1016/j.cofs.2017.09.012)